# The Protein Corona on Nanoparticles for Tumor Targeting in Prostate Cancer—A Review of the Literature and Experimental Trial Protocol

**DOI:** 10.3390/biology13121024

**Published:** 2024-12-07

**Authors:** Giulio Bevilacqua, Roberta Corvino, Anna Laura Capriotti, Carmela Maria Montone, Martina Moriconi, Stefano Salciccia, Valentina Brunelli, Valerio Santarelli, Beatrice Sciarra, Aldo Laganà, Daniele Santini, Alessandro Sciarra, Alessandro Gentilucci

**Affiliations:** 1Department “Materno Infantile e Scienze Urologiche”, University Sapienza, 00161 Rome, Italy; giulio.bevilacqua@uniroma1.it (G.B.); roberta.corvino@uniroma1.it (R.C.); martina.moriconi@uniroma1.it (M.M.); stefano.salciccia@uniroma1.it (S.S.); valentinamaria.brunelli@uniroma1.it (V.B.); va.santarelli@uniroma1.it (V.S.); alegenti@yahoo.com (A.G.); 2Department of Chemistry, University Sapienza, 00161 Rome, Italy; annalaura.capriotti@uniroma1.it (A.L.C.); carmelamaria.montone@uniroma1.it (C.M.M.); aldo.lagana@uniroma1.it (A.L.); 3Department of Pharmaceutic Chemistry, University Sapienza, 00161 Rome, Italy; beatrice.sciarra@uniroma1.it; 4Department of Oncology, University Sapienza, 00161 Rome, Italy; daniele.santini@uniroma1.it

**Keywords:** protein corona, nanoparticles, prostate neoplasm

## Abstract

Recent research suggests that once nanoparticles (NPs) come into contact with the biological fluid of cancer patients, they are covered by proteins, forming a “protein corona” composed of hundreds of plasma proteins. The concept of a personalized, disease-specific protein corona, demonstrating substantial differences in NP corona profiles between patients with and without cancer, has been introduced. Our experimental design makes a significant contribution to the field of specific protein corona nanoparticles as molecular biomarkers differentiating between benign and malignant prostatic pathology. Above all, these can be used as an indicator of the response to systemic therapy for metastatic PCa in a sector where the therapeutic choices are numerous and require increasingly valid precision medicine.

## 1. Protein Corona Nanoparticles: Definition and Mechanism of Action

### 1.1. How Protein Corona Nanoparticles Are Defined

According to the International Organization for Standardization (ISO), nanoparticles (NPs) are nano-objects with a similar length of external dimensions and global sizing around 1 to 100 nm. They differ in shape, size, and structure, being spherical, cylindrical, conical, tubular, and irregular [1].

NPs have unique properties compared to the larger dimensions of the same materials in terms of mechanical, thermal, magnetic, catalytic, electronic, and optical properties; this makes NPs suitable for many applications. They are classified into different groups by their composition: organic, carbon-based, and inorganic NPs. Organic NPs are made of proteins, carbohydrates, lipids, polymers, and other organic matrices: examples include liposomes, micelles, exosomes, dendrimers, and protein complexes. They are bio-degradable and sensitive to thermal and electromagnetic radiation; these features make organic NPs suitable for many biomedical applications such as target drug delivery and cancer therapy [1].

Carbon-based NPs are carbon atom complexes, such as fullerenes, carbon black NPs, and carbon quantum dots. These NPs have unique optical–thermal–sorption properties, electrical conductivity, high strength, and electron affinity due to sp2-hybridized carbon bonds: their applications include drug delivery, energy storage, bioimaging, photovoltaic devices, and environmental sensing [1].

Inorganic NPs include the rest of the NPs, typically metal, ceramic, and semiconductor NPs: this category comprises materials with different physical, chemical, and biochemical properties, so inorganic NPs are important in material and device creation [1].

NPs, in general, can be found in various locations within the body. Naturally occurring NPs may be biological molecules (i.e., proteins and DNA), exosomes, or environmental particles (entering the body through inhalation, ingestion, or skin contact). The greatest accumulations of NPs are found in the blood, liver, and spleen; the greater the NP, the faster it accumulates [2].

In particular, exosomes are extracellular nanovesicles secreted by all mammalian cells either in physiological or pathological conditions; they are carriers of different biomolecules such as lipids, proteins, and nucleic acids, representing extracellular messengers through intercellular communication and process modulation [3].

In vivo, in different conditions, NPs interact with bodily fluids so that a high level of proteins surrounds their surface and creates a complex, variable, and dynamic coating called a “protein corona” [4]. According to different adsorption theories, this process is energetically favorable due to a rather higher net binding energy compared to the surrounding environment, increasing the NP size by 3–35 nm and the NP surface charge to 10–20 mV [5]. The corona profile is shaped by various ambient factors, including the biomolecule concentration, pH, ionic strength, temperature, and incubation time. Once the corona forms, the nanoparticle’s identity undergoes a transformation, altering its physicochemical properties, targeting capabilities, and biological responses. These changes arise from the new biological identity imparted by the adsorbed proteins

The protein corona is composed of an inner layer of tightly bound/stronger affinity proteins with a longer lifetime (“hard corona”, HC) and an outer layer of weakly bound proteins with a shorter lifetime (“soft corona”, SC). HCs are more easily isolated and characterized: examples of NPs with HCs include liposomes, quantum dots, metallic NPs, silica NPs, polymeric NPs, and 2D materials [5]. HCs form gradually, are more stable, and comprise “relatively immobile” proteins with strong affinities for NP surfaces and low tendencies to dissociate. Soft coronas, consisting of biomolecules like proteins with low affinities for NP surfaces, undergo dynamic and reversible changes influenced by the conditions of the surrounding biological fluid. Understanding both soft and hard coronas is essential for interpreting NP stability, functionality, and interactions with biological systems. Under thermodynamically favorable conditions, proteins bind competitively to NP surfaces, forming transient NP–protein complexes composed of soft and hard corona components. Due to the high dissociation rate of soft corona proteins, current knowledge of the biological identity of the protein corona is mostly restricted to hard corona proteins (Figure 1) [5].

### 1.2. Mechanism of Action

NPs can interact with the body throughout skin and oral penetration or inhalation, subsequently moving to other body regions [1]. Once they interact with body fluids, they form the NP–protein corona complex.

Despite protein adsorption on NPs being an almost instantaneous mechanism, it is also a dynamic and continuous process, driven by unceasing protein adsorption and exchange on the corona surface: blood composition is constantly changing in time due to convection and cellular metabolism, causing the protein corona composition to vary continuously [4].

When an NP interacts with a cell membrane, it forms a very heterogeneous NP–cell interface which may influence intercellular interactions, cellular uptake, biodistribution, and immunogenicity throughout the body. This process is typically governed by the physiochemical identity of the NPs, the exposure time, and local thermodynamic exchanges [4].

More specifically, the adhesion forces can derive from either specific interactions (such as the recognition and binding of ligands on the NP surface coating to complementary receptors on the cell membrane) or nonspecific interactions (driven by pH and ionic strength, resulting, in general, in attraction or repulsion between molecules), or both [6].

An example of a specific mechanism may be protein coronas expressing opsonins, immunoglobulin G (IgG), and immunoglobulin M (IgM) on their surfaces, which can lead to different patterns of recognition, phagocytosis, circulation, and internalization processes [4]. The accumulation of proteins and subsequent formation of a corona are strongly influenced by the physicochemical properties of NPs. Key factors such as size, shape, and surface charge significantly affect the composition of the protein corona. For instance, increased surface curvature appears to help globular proteins, such as bovine serum albumin, maintain their original conformation, as observed with monodisperse silica spheres. In contrast, fibrillar proteins like fibrinogen may lose their native structure as particle size decreases. Beyond size and curvature, surface charge is another critical determinant. Moreover, potential differences in protein corona could be influenced by NPs’ administration via different routes (e.g., oral, inhalation, topical administration, intramuscular administration, and intravenous injection). Nanoparticles (NPs) can be delivered through various routes, including intravenous or intradermal injection, oral administration, or inhalation. As they traverse the body, NPs encounter dynamic biological microenvironments with varying protein compositions and concentrations, which influence corona formation and may significantly impact physiological responses. This highlights the importance of investigating how these environmental factors affect NPs, thus supporting the successful development and introduction of new nanoparticulate drug delivery systems.

## 2. How to Detect Them: Methods for Analysis

### 2.1. Main Technologies Analyzing Protein Coronas–Nanoparticles

Many techniques are employed to characterize all protein–NP interactions due to the complexity of the analysis, which involves heterogeneous features (such as size distribution, density, composition, and molecular weight) [7].

Reproducible methods should be used, and particles should be isolated without losing the attached proteins. The main methods for NP–protein corona isolation are centrifugation, magnetism, and chromatography [8]. Various organic matrices—such as blood, blood serum, plasma, and complex protein mixtures—are employed, with approaches differing based on parameters such as time, concentration, and temperature. The choice of separation method is critical and must align with the specific investigated particle type to ensure reliable outcomes. Moreover, incorporating appropriate controls is essential to prevent misinterpretation and minimize the risk of false positives or negatives, thereby ensuring robust and accurate protein corona analysis.

Centrifugation is the most common way to separate particles from a matrix. It requires precise optimization of speed, duration, and washing steps tailored to the specific characteristics of the particles to prevent undesired precipitation. At speeds exceeding 100.000 g, this technique is known as ultracentrifugation. Ultracentrifugation is mostly used for lower-density particles, and it can be either analytical or preparative. [8]. Analytical ultracentrifugation allows for real-time monitoring of the concentration of an analyte in the sample, such as fluorescent nanoparticles. The primary challenge of this method lies in the risk of false positives. Proteins or protein complexes that were not originally bound to the particle, as well as proteins that bind indirectly via particle-associated proteins, may co-sediment with the particles and their coronas during centrifugation. Conversely, false negatives can occur if proteins dissociate from the nanoparticle–corona complex under the applied centrifugation forces. To address these issues, it is crucial to carefully optimize the number of washing steps, centrifugation speed, and duration to suit the specific type of nanoparticle–corona complex and the protein-rich medium under analysis. However, such optimization is often overlooked or inadequately reported in relevant publications. Proper adjustment of these parameters is essential to ensure effective separation of nanoparticles with their corona from the surrounding medium while preventing protein aggregates from being unintentionally included in the pellet.

Magnetism is the second most used method, exploiting magnetic forces and resulting in an easier and faster method of separation. Iron oxide imparts magnetic properties to NPs, and coating with additional materials can enhance these properties. Compared to centrifugation, this method has the benefit of causing less impact on NP–corona structures. With magnetism, false positive rates are reduced due to aggregation under centrifugal forces, and the loss of proteins after multiple washing steps is minimized [8]. Iron oxide nanoparticles hold great potential in targeted cancer diagnostics. They are also available in hybrid forms, coated with materials like silica, which combine the distinct properties of both components. Magnetic separation is considered less disruptive to the structure of the nanoparticle–protein corona complex compared to centrifugation. However, it is important to note that the risk of particle agglomeration rises with increasing particle size. As a result, magnetic separation is generally not recommended for nanoparticles with diameters exceeding 10 nm. For these larger particles, a multi-step purification approach involving progressively higher centrifugation intensities may be more suitable.

Chromatography provides an investigation on the association/dissociation rates and affinity of individual proteins bound to NPs, also allowing for the collection of different fractions of a sample with less perturbation to particle–protein complexes. One common method is size exclusion chromatography, which is based on separation by the hydrodynamic volume of the analyte. This way, smaller particles interact more with the stationary phase while bigger ones move faster. A second method is flow-field-flow fractionation, such as asymmetric flow-field-flow fractionation (A4F). This technique involves the separation of analytes in a wide size range, thus reducing potential nonspecific interactions. It adopts a liquid flow established in a channel with a nonporous and a porous wall, so that particles are exposed to both a laminar flow (pushing along the tube) and a cross flow (leading to the bottom of the channel). It is used especially to analyze complex samples and stable protein coronas, with minimal perturbation [8].

### 2.2. Materials and Substrates for Analysis

In corona analytics, blood is the most studied biological fluid, especially human plasma or serum. Additionally, blood derived from common test animal species (i.e., rats, mice, or bovines) is often used in studies with animal trials or as a comparator to human blood [8].

Another matrix of study is buffers with individual or mixed proteins, investigating the specific binding behavior of a target protein (i.e., albumin) [8].

Other biological matrices are cell culture medium with blood serum, cell and tissue homogenate, and lung and nasal fluids. Moreover, few studies dealing with gastrointestinal fluids, food components, urine, or the lymphatic system are reported in the literature. Anyway, ongoing research is exploring comparisons of protein corona formation after incubation with different matrices [8].

### 2.3. Limits of Technologies

Nanomaterials possess unique characteristics, making the selection of a separation method for isolating protein–corona complexes highly dependent on the particle’s physicochemical properties, the surrounding matrix, potential unintended interactions, and the desired outcome for the corona. While centrifugation is the most commonly used technique and often the first choice for many particles, it may not be suitable for low-density analytes. Despite centrifugation being straightforward to perform, proper controls are essential to minimize the risk of false positives and negatives in protein identification. Additionally, repeated washing and centrifugation steps can lead to the loss of particle fractions, which must be carefully managed.

There are many risks of false positives using centrifugation (proteins, protein complexes originally not bound to the particle, or proteins that bind to particle-attached proteins but not to the particle itself, may sediment during the process together with the particles and their corona), and of false negatives (centrifugation forces may lead to the dissociation of proteins from the NP–corona complex). Generally, one centrifugation step is not sufficient [8]. In addition, ultracentrifugation yields false positives or negatives due to dissociation of proteins caused by centrifugal forces.

Through magnetism, the risk of agglomeration increases with particle size. In fact, its use is not recommended for NPs with diameters greater than 10 nm. Other disadvantages are represented by the limited number of suitable particle species and the possibility of interaction between magnetic particles and other required methods [8].

Despite its potential, chromatography is a more time-consuming and cost-intensive approach, making it the least frequently used method. It allows only a low throughput of samples at time, and many particles are not suitable (especially bigger sizes, polydisperse molecules, and particles adherent to the column material). Moreover, establishment methods are necessary, which require significant time and effort, further limiting its widespread application [8]. The advantages and limitations of each methodology are summarized in Table 1.

## 3. Potential Diagnostic Role in Oncology

Early cancer detection methods are needed, as currently available tests can detect only a small fraction of potential biomarkers. Recently, significant advancements have been made in proteomic analysis, with nanotechnology-based platforms emerging as promising tools [9].

Nowadays, it is well known that the protein pattern in the blood of cancer patients differs from that of healthy donors, and the molecular composition of the protein corona around NPs could change between patients with and without cancer, although the exact causes are not clearly defined. Hundreds of plasma proteins are differentially expressed, whether increased or decreased, as a cause or consequence of cancer. Consequently, the identity of NPs might be affected by protein alteration in the blood of cancer patients [10]. Researchers demonstrated that the protein corona composition could also be influenced by tumor size and the presence of distant metastases [11]. Routinary blood tests are not able to characterize each profile. The NPs’ corona characterization could detect minor changes in protein concentration either in early cancer stages or even after primary treatment (chemotherapy, surgery, etc.) [9]. A novel platform has been developed for NPs’ protein corona detection, combining the basic ideas of disease-specific protein coronas with sensor array technology. This platform consists of three cross-reactive liposomes with systematic changes in surface charge, incubated with plasma collected from diagnosed cancer patients. The resulting coronas are subsequently thoroughly characterized by several methods [11].

Different examples regarding the potential role of protein coronas–NPs as biomarkers for neoplasms are available in the literature.

PEGylated polystyrene nanoparticles (PEG-PNs) are widely used to evaluate the influence of protein corona composition in patients with non-small cell lung cancer (NSCLC) [12]. In 2022, Xu W et al. used PEG-PNs to evaluate the influence of protein corona composition in patients with NSCLC and type 2 diabetes mellitus (T2DM) compared with that of patients with NSCLC alone [12]. They collected human plasma samples which were isolated by centrifugation from heparinized venous blood, obtained from the two groups of patients. Then, fluorescent labeled polystyrene NPs were synthesized by microemulsion polymerization and subsequently modified with PEG and transferrin (Tf) to create PEG-NPs and Tf-NPs. These PEG-NPs and Tf-NPs were cultured with plasma from patients in the NSCLC with comorbidity group to obtain derived protein coronas–NPs. Thus, different NPs’ coronas were characterized using nano-liquid chromatography–tandem mass spectrometry. PEG-NPs and Tf-NPs were intravenously injected into mice with NSCLC and mice with NSCLC comorbid with T2DM to study NPs’ behavior in vivo using fluorescence imaging. The way the state of the disease affected NPs’ distribution was then evaluated: NPs accumulated in the tumor tissue at 1 h post-injection and remained in the tumor area for 24 h. In particular, Tf-NPs exhibited higher accumulation in tumor tissue than PEG-NPs and higher accumulation in the NSCLC comorbid with T2DM group than in the NSCLC group. Furthermore, researchers investigated the cellular uptake of various types of protein-coated NPs in A549 cells (a model for NSCLC), including bare NPs (PEG- and Tf-NPs), those coated with human plasma-derived proteins from NSCLC patients, and those from NSCLC patients with T2DM. A549 cells showed a time-dependent increase in NP uptake: Tf-NPs had higher uptake than PEG-NPs at 1 h and 2 h (*p* < 0.01). Pre-treatment with free Tf reduced Tf-NP uptake, suggesting that Tf-receptor-mediated endocytosis is the likely pathway for uptake. Tf-NP-coated coronas from individuals with NSCLC comorbid with T2DM showed greater accumulation in A549 cells. This study highlights how the comorbidity clinical status of patients influenced the protein corona composition, thereby affecting the bio-behavior of NPs [12]. Notably, Tf-NPs demonstrated greater accumulation in NSCLC groups comorbid with T2DM, providing insight into how the disease state plays a pivotal role in determining the distribution of nanoparticles. Furthermore, the interaction between cellular receptors and nanoparticle surface ligands is crucial in shaping the biological fate of nanoparticles. NSCLC comorbid with T2DM-derived protein coronas contain more fibrin and polyproteins, which may help nanoparticles evade non-characteristic intake [12].

In breast cancer, rare-earth-doped nanoparticle (RENP)–cell interactions were analyzed at the earliest stage of tumor development, showing how the protein corona coating of RENPs determines the unique pathways by which RENPs accumulate in cancer cells [13]. Voronovic et al. investigated the corona composition and its impact on the cellular uptake of citrate-, silica-, and phospholipid micelle-coated RENPs, selecting two cell lines (MDA-MB-231 and MCF-7, as breast cancer model systems) and utilizing confocal fluorescence microscopy to evaluate their emission intensity after incubation. It was stated that the protein corona around RENPs may play a major role in their stability, accumulation dynamics, and cellular uptake mechanisms, as MDA-MB-231 cells accumulated RENPs in greater quantities than the MCF-7 cell line, mostly citrate-coated ones. In addition, they found that proteins located in RENP coronas may activate the mechanism of micropinocytosis in both breast cancer cell lines [13].

Another study demonstrated how the hard corona (HC) formed on lipid NPs after pancreatic cancer blood exposure differs from that of control blood in terms of the major protein bands expressed [9]. In 2016, Caputo et al. researched new applications of nano-bio interactions to find new diagnostic approaches for pancreatic cancer. Twenty tumor and five non-tumor blood samples were collected to interact with designed lipid NPs, so that specific protein coronas may coat the NPs. These patterns were isolated and then analyzed using SDS-PAGE. The results showed that protein coronas of pancreatic cancer patients were more enriched than the control ones, with a discrimination rate of 88% [9].

There were also findings concerning the relationship between the expressed corona molecular weight and the pancreatic cancer stage according to the TNM classification. These findings suggest that size may be the main factor determining tumor prognosis, since it reflects tumor biology [14]. Caputo et al. aimed to investigate how pancreatic ductal adenocarcinoma tumor size and the presence of distant metastases influence the protein corona composition. The authors collected 20 tumor blood samples, exposed them to lipid NPs, causing an interaction, and then characterized molecules using SDS-PAGE. The results allowed them to distinguish T1-T2 (according to the TNM staging system) cases from T3 and, above all, from metastatic ones (*p* < 0.05), in particular due to the differences in molecular weight (25–50 and 50–120 kDa) [15].

### Key Points Summary

The molecular composition of the protein corona around NPs can change between patients with and without cancer, and it can be influenced by tumor size and distant metastases. The NPs’ corona characterization could detect minor changes in protein concentration, not only in early cancer stages, but also after primary treatment. In the literature, research is mainly focused on lung, breast, and pancreatic cancer.

## 4. Potential Therapeutic Role in Oncology

Therapeutics based on NPs have been widely explored, resulting in many applications in clinical practice today. In the oncology therapeutic field, most studies focus on the use of NPs rather than on the potential role of the protein corona.

Recently, many studies have focused on the potential therapeutic role of NPs, particularly as drug delivery vectors. Research indicates that there are general requirements for an effective system for cancer treatments: NPs must be biocompatible, highly bioavailable, and stable under physiological conditions. They should be able to target only tumor cells without interfering with surrounding healthy cells and to release their load as soon as they reach the target site [15].

NPs’ size is the most influential factor for adequate biodistribution and effective drug delivery. Thanks to their relatively small size, NPs are very efficient at crossing membrane pores. Tumors have leaky vasculature, letting NPs easily penetrate small tumor vessels, while normal blood vessels typically prevent NP extravasation, thus avoiding agglomeration in other parts of the body. In general, smaller-sized particles (<50 nm) have better antitumoral efficiency. However, as different organs have different size uptake characteristics, research is focusing on this concept for the development of a more organ-specific cancer treatment [15,16].

In addition to size, NPs’ shape is a critical factor influencing interactions between molecules, controlling fluid dynamics and affecting cellular uptake. The reticuloendothelial system (RES) is the main site of nanomedicine storage, impacting the toxicity and immunogenicity of injected drugs [16]. The nano-drug delivery system undergoes three key steps in order to effectively reach the tumor. First, the nanoparticles circulate throughout the body, where they are partially engulfed by macrophages in the reticuloendothelial system. Next, they penetrate the tumor site and release the drugs. Finally, they accumulate in the tumor and exert therapeutic effects. However, before reaching the tumor, nanoparticles face several challenges, such as recognition as foreign bodies and phagocytosis by macrophages. These challenges can complicate the use of nanoparticles for tumor treatment, with only a low rate of nanoparticles ultimately reaching the tumor site. The protein coronas’ composition and interaction with NPs can help them escape the RES and macrophages’ interaction, ultimately improving drug delivery and transportation [16].

The complexity of the NP surface leads to varied interactions, degradation, agglomeration rates, and cellular uptake. Consequently, the protein corona–NP complex plays a major role in the biodistribution–biocirculation–biocompatibility of potential carried drugs. Recent research demonstrated that different surface properties may be necessary depending on cancer type and stage. To date, many strategies have been developed to assist tumor uptake of NPs. These strategies include manipulating NPs’ design (influencing tumor infiltration and/or retention of drugs), modifying the tumor microenvironment by the coadjutant retention of drugs, and modifying the tumor microenvironment by coadjutant treatments such as photodynamic therapy, radiotherapy, and immunotherapy [4].

Applications in clinical practice may be the utilization of gold NPs because of their optical and tunable properties, with an easy modifiability due to their negative surface charge. For instance, Methotrexate (MTX) conjugated with gold NPs has higher cytotoxicity than MTX alone. Doxorubicin (DOX) has higher potential against the multidrug resistant MCF-7/ADR breast cancer cell line if conjugated with gold NPs. Other examples may include the conjugation of gold NPs with peptide–drug conjugates (PDCs) and phytochemicals such as kaempferol [17,18]. In 2007, Chen YHP et al. [18] proposed a new MTX formulation (MTX-AuNP conjugate) to prolong drug retention in tumor cells and alter its pharmacokinetic behavior. Spectroscopic examinations were conducted revealing how MTX accumulation is faster and higher in tumor cells treated with MTX-AuNP than those treated with free MTX. Moreover, MTX-AuNP showed higher cytotoxic efficacy on several tumor cell lines when compared with an equal dose of free MTX. In a recent study on mouse models of ascites Lewis lung carcinoma (LL2), the administration of MTX-AuNP rather than an equal dose of free MTX significantly (*p* = 0.0041) increased tumor growth suppression [15].

NPs have the ability to overcome solubility and stability problems of anticancer drugs by encapsulating the compound within a hydrophilic nanocarrier. When perishable, drugs can also be paired with synthetic nanocarriers. NPs’ physicochemical properties enhance drug penetration and redirection, allowing passive or active targeting in a selective manner. Additionally, nanocarriers can influence the circulation time of a drug by expelling upon contact with specific environmental factors (such as pH), resulting in a stimuli-sensitive treatment [17].

An example of targeting is represented by the optimization of anti-HER2 monoclonal antibodies, creating “nanobodies”: they consist of the antigen-binding domain of the heavy chain-only camelid antibody, showing greater stability, assembled through covalent binding [17]. In 2017, D’Hollander et al. proposed a chemical strategy to overcome protein corona-targeting issues in in vivo biological systems by reducing their thickness. Nanobodies were covalently bound to AuNPs through a self-assembled monolayer interface. They found an optimal blocking agent (2-mercapto ethanol), interfering with the active groups of the self-assembled monolayer on AuNPs. Two cell lines were used to test this approach both in vitro and in vivo in a mouse model: an ovarian cancer cell line (SKOV3) showing high HER2-receptor expression and a hamster ovarian cancer cell line (CHO) as a negative control. Through darkfield microscopy and photoacoustic imaging, it was seen how a reduction in the protein corona size increased the specificity of the functionalized molecules towards HER2-expressing tumor cells [17].

Immunotherapy may benefit from these findings, as the protein corona coating on NPs’ surface could modulate immune response. On the one hand, precoated molecules, such as liposomes, may reduce immunogenicity. On the other hand, protein coronas can act as ligands of immune receptors, causing a cytokine storm and stimulating macrophages to produce a pro-inflammatory response [18]. Cai R. et al. discovered that interleukin-1β (IL-1β) levels are positively correlated with the abundance of immune-related proteins within the protein corona coating [19]. Since the surface chemistry-induced specific protein corona affects the phagocytosis and immune responses of macrophages, it can influence the internalization pathways and cytokine secretion profiles of macrophages. The release of interleukin-1β (IL-1β) by macrophages is directly dependent on the number of proteins involved in immune responses (e.g., acute phase, complement, and tissue leakage proteins) present in the acquired nanoparticle corona [20].

### Key Points Summary

Few studies have focused on the potential therapeutic role of NPs, especially on their possible use as vectors for drug delivery. NPs designed for this purpose must be biocompatible, highly bioavailable, and stable under physiological conditions. They should be able to target only tumor cells without interfering with surrounding healthy cells and to release their load as soon as they reach the target site. NPs’ size is the most influential factor in biodistribution and delivery. Their role in optimizing monoclonal antibody, immunotherapy, and chemotherapy targeting has also been investigated.

## 5. Advances in Prostate Cancer

Currently, there are few studies in the existing literature specifically focused on the applications of NPs in prostate cancer (PCa) treatment, either as diagnostic or therapeutic tools.

### 5.1. Diagnostic Applications

Recently, the concepts of protein coronas, sensor arrays, and supervised classifiers were merged to create the concept of a “protein corona sensor array”, aimed at effectively identifying and distinguishing diseases.

Digiacomo L. et al. [20], in 2020, established the experimental validation capabilities of the protein corona sensor network in oncology and neurodegenerative disorders. This research assesses the viability of identifying breast and prostate cancers using the protein corona sensor array platform. To achieve this goal, they utilized three cross-reactive liposomal formulations with unique physicochemical characteristics, leading to varying affinities for specific plasma proteins. This strategy can significantly enhance the quantity and diversity of plasma proteins potentially linked to cancer. By utilizing arrays of NPs with unique physiochemical traits, distinct protein corona profiles can be produced. This indicates that a single NP type’s protein corona composition could yield unique ‘fingerprints’ for each condition. In this investigation, three liposomal formulations with differing lipid compositions and surface charges were exposed to human plasma from patients diagnosed with two prevalent cancer types, breast cancer and PCa, alongside a control group of healthy donors. Instead of targeting a specific biomarker, they examined alterations in the protein profiles that facilitate the differentiation between patients with and without cancer. Overall, the size, zeta potential, and nano-liquid chromatography–tandem mass spectrometry findings suggest that the liposome corona is influenced by the liposomes’ surface chemistry and varies across different cancer types. Utilizing statistical methods, they were ranked based on their capability to distinguish between individuals with and without cancer. Significantly, the proteins demonstrating the highest discrimination potential were clearly linked to various cancer aspects. One important process in tumor development is angiogenesis, the physiological mechanism through which new blood and lymph vessels arise from pre-existing vessels, supplying tumors with necessary nutrients and cytokines for the growth and systemic spread of cancer cells. The coronas of cancer patients revealed several proteins associated with angiogenesis. The coronas from cancer patients showed substantial differences from those of healthy individuals in its levels of FBLN1, which belongs to the fibulin family, a class of proteins involved in organizing the stromal matrix. The results showed a clear separation of cancer patients and control subjects in two-dimensional spaces with axes defined by corona proteins with the highest discrimination ability. Following incubation with HP from healthy individuals, 178, 256, and 179 proteins were identified in the coronas of L1−HP, L2–HP, and L3–HP complexes, respectively. The percentage of proteins common to all three formulations spanned from ≈50% (L2–HP complexes) to ≈72% (L1–HP and L3–HP complexes). However, unique proteins comprised a minor fraction of the coronas, with the coverage percentage ranging from ≈7% (L1–HP complexes) to ≈29% (L2–HP complexes). After their interaction with HP from breast cancer and prostate cancer patients, similar results were obtained. Coronas from cancer patients and healthy individuals were compared and RPA values of breast cancer and control groups were statistically different in 172 proteins (L1: *n* = 62; L2: *n* = 71; and L3: *n* = 39). This number was 157 when the coronas of PCa patients and control subjects were compared (L1: *n* = 55; L2: *n* = 58; and L3: *n* = 44). This characterization has potential for improving the early detection of cancer using a simple blood test [19].

Huo Qu et al. [21] in 2011 developed an NP immunoassay for serum protein biomarker detection and analysis in which a serum sample was first mixed with a citrate-protected AuNP solution. Proteins from the serum were adsorbed to the AuNPs to form a protein corona on the NPs’ surface. An antibody solution was then added to the assay solution to analyze the target proteins of interest that were present in the protein corona. The protein corona formation and the subsequent binding of antibodies to the target proteins in the protein corona were detected by dynamic light scattering (DLS). They discovered multiple molecular aberrations associated with PCA from mouse and human blood serum samples. From the mouse serum study, they observed differences in the size of the protein corona and mouse IgG level between different mouse groups. The experiment enclosed the mouse implantation of the rapidly proliferating prostate cancer cell line PC3, the slowly progressing tumor cell line LnCaP, and a third group of mice treated with phosphate-buffered saline (PBS) solution as a control. Mice implanted with PC3 cells developed significantly larger tumors (measured in grams) than those injected with LnCaP cells (weighing in tens to hundreds of milligrams). The average weight ratio of the tumor over body weight was approximately 5% for the PC3 mice and less than 0.3% for the LnCaP mice. The average particle size increase in the healthy control group was 75 nm, significantly higher than the PC3 mice with an average particle size increase of 24 nm and an average particle size increase for LnCaP mice of 43 nm. It was also noticed that within the healthy control and the LnCaP mouse group, there were substantial variations between individual mice: the particle size increase varied from 20 to 110 nm for the LnCaP mice and from 40 to 120 nm for the healthy control mice with no prostate tumors in the control mice. Three groups of human serum samples were included in this study: normal healthy donors (*n* = 15); patients diagnosed with BPH (*n* = 10); and patients diagnosed with PCa with stages from T1c to T3b (*n* = 25). It was found from both the mouse model and the human serum sample study that the level of vascular endothelial growth factor (VEGF) adsorbed to the AuNPs was inferior in cancer samples compared to non-cancerous or less malignant cancer samples. This research demonstrates a notable disparity in the “size” of the serum protein corona that is formed on the surface of AuNPs between cancerous and non-cancerous or less aggressive tumor-bearing mouse models [21].

Ahmadianpour et al. [22], in 2020, studied the application of AuNPs for the early detection of PCa. In this study, blood samples of 60 male subjects aged 40–90 years were collected from 20 healthy individuals, 20 patients with BPH, and 20 patients with PCa. Optical scattering changes were measured by the level of AuNPs mixed with different sera, and the responses were compared with the PSA (Prostate-Specific Antigen) index of the subjects. No significant differences were found in the size of the corona protein structure between the three groups of males with PCa or BPH and healthy males. No correlation was found between the DLS concentration and PSA serum level due to changes in ambient temperature, prolonged test duration, or high IgG levels in apparently healthy individuals. They ascertained that DLS has major limitations for PCa detection, so it cannot be a simple and accurate method for the early detection of this tumor [22].

Lately, researchers have focused on tumor-linked exosomes through liquid biopsy evaluation, obtaining PSA exosomes with more sensitivity and specificity than traditional PSA blood analysis and representing a promising biomarker [3].

Logozzi et al. demonstrated how analyzing PSA-expressing exosomes can differentiate between healthy patients and those with prostate disease, also discerning into benign and malignant pathology. The study enrolled 240 male patients, of which 80 were controls, 80 had BPH, and 80 had PCa. Exosomes were extracted from an EDTA-treated blood sample using centrifugation; then, they were qualified and quantified using different methods. In particular, the IC-ELISA method had a 98.57% sensitivity and an 80.28% specificity in discriminating malignant from benign prostatic pathology. Also, combining IC-ELISA and NFSC led to an increase of up to 96% in sensitivity and 100% in specificity. Moreover, a significant (*p* < 0.0001) increase in the number of exosomes and a lower size in prostate cancer patients was demonstrated [3].

### 5.2. Therapeutic Applications

An ongoing clinical trial, active from 2020, is AuroLase^®^ NCT04240639, which provides ultra-focused tissue ablation therapy for solid tumors and aims to enhance treatment effectiveness while reducing the side effects commonly linked to surgery, radiation, and conventional focal therapies [23,24]. It employs the “optical tunability” of a novel class of NPs known as AuroShells, which absorb near-infrared wavelengths of light that harmlessly infiltrate human tissue and expose the tumor to a near-infrared laser. The NPs selectively seize the photonic laser energy, transforming the light into heat, effectively obliterating the tumor and its nourishing blood vessels while preserving adjacent tissues. AuroLase Therapy utilizes an FDA-approved laser that produces near-infrared energy within clinically defined parameters (output, duty cycle, duration) and incorporates an FDA-sanctioned fiber optic probe for percutaneous energy delivery. AuroShell particles (also referred to as “nanoshells”) feature a gold metal shell surrounding a non-conductive silica core, functioning as the external absorber of the near-infrared laser energy channeled by the probe. These AuroShells are introduced intravenously and, due to their diminutive size, they can gather in the tumor through its porous vasculature. NPs cannot penetrate normal blood vessels; hence, they do not accumulate in healthy tissues. Once concentrated in the tumor, the region is targeted with a near-infrared laser at carefully selected wavelengths to ensure maximum light penetration through tissues. The AuroShells are engineered to absorb this specific wavelength and convert the photonic laser energy into sufficient heat to ablate the tumor [24]. Nanospectra’s proprietary nanoshells navigate freely in the bloodstream and converge in the tumor. Utilizing cutting-edge imaging technology, the clinician precisely locates the prostatic lesion and positions the optical fiber probe through targeted MRI–ultrasound fusion technology on the prostate gland. PCa tissue could be ablated while preserving adjacent healthy tissue. Currently, AuroShell particles are investigational and can only be accessed through designated, FDA-authorized clinical study sites [24].

### 5.3. Key Points Summary

In the field of prostate cancer, research is in its initial phase. In the context of diagnostics, the use of PSA-expressing exosomes showed potentiality in better discrimination between PCa and BPH diseases. In the context of therapies, support for ultra-focused treatments and targeting appears to have the greatest application possibilities.

## 6. Experimental Trial: Protocol Design

### 6.1. Premise

Despite advances in genomics, proteomics, metabolomics, and lipidomics, to date there is still an absence of predictive protein and metabolic biomarkers. Proteomics and metabolomics approaches, based on mass spectrometry techniques, are undoubtedly of high interest but face challenges such as low sensitivity and specificity due to the low concentration levels of biomarkers in human plasma. Clinical application is further hampered by factors such as sample size, intra-individual variability, the presence of bias, and overall cost. These challenges highlight the complexity of using proteomic and metabolomic approaches for effective detection and treatment of PCa. The National Cancer Institute (NCI) recognizes the potential of technologies based on the use of nanoparticles in revolutionizing clinical approaches in the diagnosis and prognosis of cancer. Recent research suggests that once nanoparticles (NPs) come into contact with the biological fluid of cancer patients, they are covered by proteins, forming a “protein corona” composed of hundreds of plasma proteins. Hajipour and colleagues [25] introduced the concept of a personalized, disease-specific protein corona, demonstrating substantial differences in NP corona profiles between patients with and without cancer. As the protein corona field grew, limited interest in the metabolite corona began to emerge with investigations into the lipid composition of the corona around inhaled nanomaterials and, eventually, more holistic analyses of the metabolite corona. Chetwynd and co-workers [26] suggested that the metabolite corona coexists with the protein one since these smaller molecules can fit between proteins and are often bound into protein complexes. The metabolite corona is complementary to protein coronas, following similar rules of adsorption based on abundance and affinity, leading to metabolite fingerprints akin to protein fingerprints. Understanding how NPs interact with the metabolites in the biological milieu, particularly in cancer patients, holds promise for advancing diagnostic capabilities. The metabolite corona can provide valuable information about the individual’s metabolic profile, offering insights into the unique metabolic characteristics associated with cancer. In conjunction with the protein corona and other emerging concepts, it contributes to a more comprehensive understanding of the complex interactions between NPs and biological components. These insights may pave the way for innovative diagnostic approaches and personalized medicine strategies in cancer diagnosis and treatment.

### 6.2. Study Design

We developed the design of an experimental prospective single-center study with patients allocated in a 1:1:1 ratio of one of the three arms. A longitudinal analysis will also be applied to a 12-month interval observation. The protocol has been approved by our Ethic Committee Sapienza Policlinico Umberto I, Prot 0919/2021. The protocol will start on January 2025.

### 6.3. Endpoints

#### 6.3.1. Overall Aim

The protocol will aim to develop and implement sensitive nanotools with two distinct objectives. It will carry this out by designing nanoparticles (NPs) capable of selectively binding and detecting biomarkers in order to build a predictive diagnostic model to effectively discriminate between patient sera affected by benign prostatic hyperplasia (BPH) and prostate cancer (PCa). This initiative will be driven by improving diagnostic precision and accuracy, enabling early and accurate differentiation between these two conditions. Secondly, within the population with PCa, in cases of initial advanced metastatic diagnosis (mHSPC = metastatic hormone-sensitive prostate cancer), the objective will be to find biomarkers capable of predicting the response to systemic treatments to improve the precision and efficiency of monitoring treatment outcomes, leveraging the aforementioned nanotechnology and leading to the development of personalized therapeutic strategies for patients.

#### 6.3.2. Primary Endpoints

The first primary endpoint is the construction of predictive diagnostic models on serum from patients with a histological diagnosis of PCa versus patients with BPH using proteomics, metabolomics, and lipidomics of the biomolecular coronas that form between nanoparticles and serum.

A second endpoint is the construction of a predictive model for the response to treatment of patients with an initial diagnosis of mHSPC for the personalization of therapies and for the stratification of such patients considering individual characteristics such as genetics, lifestyle, and more, as foreseen by precision medicine, using metabolomics and lipidomics of the biomolecular coronas that form between nanoparticles and serum.

#### 6.3.3. Secondary Endpoints

A secondary endpoint is the construction of mathematical models for the correlation of proteomic, metabolomic, and lipidomic data with clinical data of the three patient populations (non-metastatic PCa, BPH, and mHSPCa).

### 6.4. Population and Eligibility Criteria

Three different populations will be considered and compared.
-Subgroup 1: Patients with initial histological diagnosis from prostate biopsy of PCa considered for radical prostatectomy as primary therapy, as a therapeutic choice shared with the doctor and in accordance with international guidelines. This treatment will follow normal clinical practice for this pathology and regular clinical and therapeutic procedure. Staging of these patients will be non-metastatic PCa at imaging (multiparametric magnetic resonance of the prostate).-Subgroup 2: Patients with initial histological diagnosis from prostate biopsy of advanced metastatic PCA (mHSPC) to be subjected to standard pharmacological treatments with androgen deprivation (ADT) and new androgen-targeted treatments (ARPI) or chemotherapy with taxanes, as a therapeutic choice shared with the doctor and in accordance with international guidelines. This treatment will follow normal clinical practice for this pathology and regular clinical and therapeutic procedure. Staging of these patients will be metastatic at imaging (PET CT scan or bone scan and CT scan).-Subgroup 3: Patients with initial diagnosis of BPH who have not undergone treatments. A healthy control population is not included but the BPH population will be used as a control. The clinical comparison will use male subjects of the same age (40–75 years) to determine the expression of a potential marker for prostatic neoplasia, involving the evaluation of differences within a population with benign prostatic pathology such as BPH and not healthy subjects (those without BPH) of mismatched age.

#### 6.4.1. Subgroup 1 and 2

Inclusion criteria: Male patients aged between 40 and 75 years, of any ethnicity, with an initial histological diagnosis of PCa from prostate biopsy. Non-metastatic versus metastatic clinical stage, intermediate or high-risk class according to d’Amico, indication as primary treatment for radical prostatectomy (Subgroup 1) or systemic therapy with androgen deprivation and ARPI or taxanes (Subgroup 2).

Exclusion criteria: Local or systemic therapies for PCa, other neoplasms in the active phase or undergoing treatment, ongoing oncological therapies (chemotherapies, target therapies, radiotherapies), hormonal or steroid therapies in progress or with drugs known to interfere with the evaluation foreseen in the study.

#### 6.4.2. Subgroup 3

Inclusion criteria: male patients aged between 40 and 75 years, of any ethnicity, with an initial diagnosis of BPH.

Exclusion criteria: local or systemic therapies for BPH, suspicion or diagnosis of prostatic neoplasia, other neoplasms in the active phase or undergoing treatment, ongoing oncological therapies (chemotherapies, target therapies), hormonal or steroid therapies in progress or with drugs known to interfere with the assessment intended for the study.

### 6.5. Methods

Under a regime of fasting from liquids and solids from midnight the previous evening, in the morning around 8.00 AM a blood sample will be taken from all patients. In Subgroup 1 and 3, the blood sample will be obtained only at baseline. In Subgroup 2, the blood sample will be obtained at baseline before the beginning of therapy and at 3-, 6-, 12-month intervals during systemic therapy. Sample storage will be performed at room temperature for 30 min, followed by centrifugation of the sample (3000× *g*; duration: 15 min) and subsequent separation to obtain a plasma sample. The plasma samples will be transported within 30 min to the chemistry laboratory in a suitable refrigerated container with ice at a temperature of 4 °C after centrifugation of the blood to remove the erythrocytes. As soon as they are delivered to the laboratory, the plasma samples appropriately labeled with the sample identification code will be stored at −80 °C until analysis.

For protein and metabolite corona experiments, in collaboration with the nanodelivery Lab of Sapienza of Rome, we developed a cross-reactive sensor array platform with cancer detection capacity made of three liposomal formulations with different surface charges, i.e., cationic 1,2-Dioleoyl-3-trimethylammonium propane (DOTAP), anionic 1,2-Dioleoyl-sn-glycerol-3-phosphoglycerol (DOPG), and a zwitterionic mixture made of dioleoylphosphatidylcholine (DOPC) and cholesterol [19]

For the proteomics–NP studies, proteins will be identified and quantified by nano-high-performance LC (nanoHPLC) coupled to MS/MS (nanoHPLC−MS/MS). The acquired raw MS/MS data files from Xcalibur software (version 2.2 SP1.48, Thermo Fisher Scientific, Washoe County, NV, USA) will be searched against the Swiss-Prot human database using the MaxQuant search engine with the automatic setting for tryptic peptide matching and label-free analysis. For each NP type, a list with all the identified proteins and their relative protein abundance (RPA, a quantitative estimation of their abundance within the protein corona) will be provided. The list of proteins will be utilized to discover new biomarkers associated with the PCa using sensor array technology [20].

Metabolites will instead be analyzed by an untargeted metabolomic approach. Before analysis, Quality Control (QC) samples will be prepared for further instrumental system conditioning and sample normalization over time. Data acquisition for samples, controls, and QCs will be obtained by ultra-high-performance liquid chromatography coupled with untargeted high-resolution mass spectrometry (UHPLC-HRMS). A data acquisition worklist will comprehend a QC run every 5 sample runs for subsequent QC-based normalization. At the end of the data acquisition worklist, a series of QCs will run in data-dependent acquisition tandem mass spectrometry (MS/MS) for subsequent tentative compound identification. All samples will be acquired in positive and negative ion mode for maximum compound coverage. After raw data collection, pre-processing analysis using Compound Discoverer software on samples, controls, and QCs will be needed to extract the *m*/*z* from the raw files, to align the variables (features) among all acquired runs, normalize each feature over time, and remove contaminants present in the blank sample. After data pre-processing, a data matrix will be obtained for further statistical analysis. Features that will be selected from statistical analysis as putative biomarkers will be then tentatively identified by matching the experimental MS/MS spectra to those present in databases, such as the Human Metabolome Database (HMDB), Lipid Maps, Kyoto Encyclopedia of Genes and Genomes (KEGG pathway) and Metabolika. Lipidomics will be carried out using a similar approach to metabolomics, but all instrumental parameters will be adjusted to lipids, including chromatography (C8 column, mobile phases, and gradient), MS tuning and parameters (*m*/*z* range and injection time), and pre-processing and identification tools (Figure 2).

### 6.6. Statistical Analysis

In the analytical field, we often work with complex problems in which the variables are numerous and sometimes uncontrollable; furthermore, information may be difficult to extract due to the presence of experimental noise, random or systematic correlations between variables, and useless or redundant variables. It is therefore essential, for a correct interpretation of the analytical data, to eliminate redundant and superfluous information to evaluate which are the most significant variables and consider any correlations between them.

For proteomics analysis, statistics and multivariate data processing will be carried out. A three-dimensional data matrix consisting of sample information, identified peptides, and the normalized ion intensity will be generated. The resultant three-dimensional matrix will be imported to SIMCA-P 14.1 software (Umetrics AB, Umeå, Sweden) for multivariate statistical analysis, including principal component analysis and orthogonal partial least squares discriminant analysis (OPLS-DA). The functional analysis of differentially expressed proteins will be performed using Gene Ontology (GO). The strength of this approach lies not only in pinpointing individual predictors (i.e., single biomarkers) but also in the pattern recognition enabled by a protein corona sensor array. The differentiation between the groups of samples may arise due to a “global change” involving several predictors that are systematically altered simultaneously, resulting in distinctive global protein patterns unique to the pathological conditions. By concentrating on unique patterns derived from a large number of subjects through a set of informative predictors, we will achieve more accurate predictions of PCa and different stratified mHSPCa patients than current methods allow.

For metabolites–coronas, a principal components analysis (PCA) and discriminant analysis (partial least squares discriminant analysis, PLS-DA) will, in general, be applied. For the purposes of the project, the first step will be the application of the unsupervised PCA technique, to understand the tendency of the samples to divide into different clusters depending on whether they belong to the group of healthy subjects or to the group of pathological subjects, and to highlight the possible presence of outliers (for example, very diluted samples). To further characterize differences between groups, supervised PLS-DA methods will be used. The application of PLS-DA could allow us to obtain a clear separation between different groups consistent with the characteristics of the pathology.

## 7. Conclusions and Future Perspectives

Compared with previous review articles, the novelty of this review is represented by the analysis of the possible clinical applications of protein corona NPs, focused on PCa, and the presentation of a new clinical protocol in the metastatic phase of PCa.

In recent years, the management of patients with PCa has undergone huge changes, leading to substantial advances in treatment outcomes. Clinical management has benefited from research results which are leading to the application of three interconnected concepts: intensification, anticipation, and precision medicine. In the different stages of PCa, we are witnessing a growing selection of patients based on the principle of precision medicine. With this approach, therapy can be tailored by identifying a population that will likely benefit from the intensification and anticipation of treatment, compared to a population in which a deintensification of care is more useful.

These concepts require parameters that allow for increasingly precise stratification of patients into risk classes, in order to identify the more adequate therapeutic choice among multiple options. The genetic analysis of the pathogenetic variants (PVs) of the homologous recombination repair (HRR) genes currently represents the most effective method for prognostic evaluation of individual patients with PCa and allows for effective precision medicine, such as the use of PARP (Poli-(ADP-ribosio)-polimerasi) inhibitors.

The research and application of nanoparticles hold high potential in the identification of predictive proteins or molecular biomarkers. In particular, the isolation of a protein corona developed by the interaction of nanoparticles with biological fluids from neoplastic cells can facilitate the identification of disease-specific systems and highlight significant differences between benign and neoplastic conditions. Consequently, the isolation of specific protein corona nanoparticles could represent a further step in the concept of precision medicine and targeted therapeutic choice in patients with PCA.

To date, available data on the application of this concept in PCa, both at the level of basic research and even more so in clinical research, are limited and in an initial phase. Currently, research is more focused on the identification of disease-specific protein corona nanoparticles with prognostic capacity and markers of response to therapy than on possible therapeutic applications. The major limitation of these studies remains the complexity of the methodology and technology required for protein corona nanoparticles’ isolation. The complex accessibility and diffusion in the case of significant research findings is an additional challenge. Technology should evolve to simplify these methodologies and reduce required time and costs, in order to make these techniques more easily accessible and widespread. Furthermore, additional clinical studies on their application in the diagnostic and therapeutic phase of specific tumors, such as Pca, are required.

Our experimental design, which we are starting to apply in a comparative and longitudinal research on PCa patients, is expected to make a significant contribution to the field of specific protein corona nanoparticles as molecular biomarkers, distinguishing between benign and malignant prostatic diseases, and above all as indicators of response to systemic therapy for metastatic PCa. A domain for the latter will require the therapeutic choices to be numerous and will require increasingly valid precision medicine.

The hypothesis is that identifying disease-specific protein corona nanoparticle systems could complement genetic analysis in the tailoring of therapeutic options for PCa patients.

## Figures and Tables

**Figure 1 biology-13-01024-f001:**
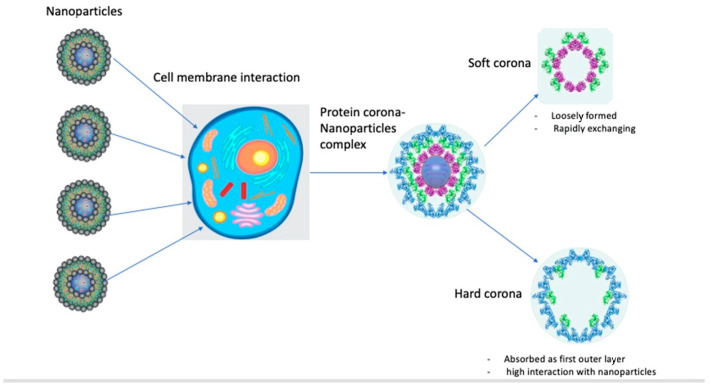
Diagram for protein corona–nanoparticle interaction.

**Figure 2 biology-13-01024-f002:**
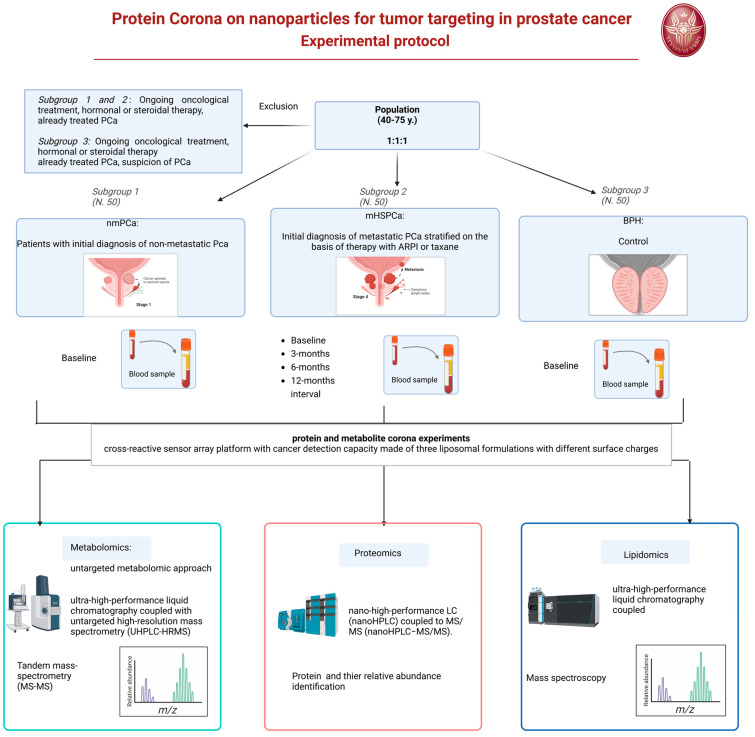
Experimental protocol flow chart.

**Table 1 biology-13-01024-t001:** Advantages and limitations of different methodologies.

Assay	Advantages	Limitations
Centrifugation	Separation according to density and size. The most used, with many applications; easy to use. High throughput. Possible optimization of centrifugal speeds and times. High resolution results. Possibility to separate coexisting populations in situ.	Long centrifugation time; false particle protein interaction. Several purification steps needed. Possible agglomeration of magnetic NPs. Outcomes influenced by centrifugation force, washing duration, and solution volumes. The smaller and less dense the NPs, the higher the chances of aggregation. Not well suited for very small (5–20 nm) or low-density NPs.
Size Exclusion Cromatog-Raphy (SEC)	Flexible, with many stationary/mobile phases. Standard lab equipment. Higher resolution and recovery. Systematic methodology. Less perturbing.	Possible interactions. Low selectivity with high molar mass analytes. Low throughput.
Asymmetric Flow-Field Flow (A4F)	Not necessary an extensive sample preparation. Reduced alteration of coronas. Investigation of heterogeneous complexes. May be coupled with several separation techniques. Possible automation. Short measurement time. Easy collection. Multifunctional. Absence of stationary phase or packaging material.	Long establishment process. Adjusted for every particle type. Low throughput. Expensive. Division into several experiments. Possible loss of samples due to adsorption. No full recovery of fraction for other experiments.
Magnetism	Low impact on NPs. High throughput.	Only small (10 nm) magnetic particles. Decrease in separation with decreasing magnetism.

## Data Availability

This is a review article without presentation of data.

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
