# Peer review of "The Protein Corona on Nanoparticles for Tumor Targeting in Prostate Cancer—A Review of the Literature and Experimental Trial Protocol"

_biology, 2024, doi:10.3390/biology13121024_

Round 1
Reviewer 1 Report
Comments and Suggestions for Authors
The current study mainly focuses on the protein corona formation on nanoparticles (NPs) when they encounter biological fluids and its implications for targeting tumors, specifically prostate cancer.
1) The article overall is a well structured one. However, while discussing the complex methodologies, concise summaries may be provided for easy understanding.
2) In the methodology for the experimental trial, a flowchart may be provided to increase the readability.
3) The current manuscript very rightfully provides the limitations of the nanoparticles. However, it would be better to suggest some ways to overcome these hurdles.
4) A diagram would have been very insightful to incorporate in the Manuscript showing the interactions between the protein corona and the nanoparticles.
Author Response
Dear Reviewer 1: Sincerely thank you for your comments. I believe with now the review is strongly improved. Changes related to Reviewer 1 comments are evidenced in red.
Comment 1. The article overall is a well structured one. However, while discussing the complex methodologies, concise summaries may be provided for easy understanding
Answer1: as requested at the end of each paragraph 3-5 a concise summary as " key points summary" has been enclosed, subparagraphs 3.1; 4.1; 5.3. Paragraph 1 is introduction and paragraph 2 on methodology has a summary in Table 1.
Comment 2: In the methodology for the experimental trial, a flowchart may be provided to increase the readability.
Answer 2: as requested, now Figure 2 is arranged as a flowchart for methodology of the experimental trial
Comment 3: The current manuscript very rightfully provides the limitations of the nanoparticles. However, it would be better to suggest some ways to overcome these hurdles.
Answer 3: In Conclusions paragraph 7 we underlined this limitation of these analyzes: "
The major limitation of these analyzes is and probably could remain soon, the complexity of the methodology and technology required for their isolation, the complex accessibility and diffusion in case of significant results coming from the research." The possible solution is not easy but it has been proposed in the same paragraph: "Technology should make an effort to simplify these methodologies, reduce times necessary and costs, so to make these techniques more easily accessible and disseminated. Furthermore, more clinical studies are needed focused on the possible application in the diagnostic and therapeutic phase, specifically on single tumors such as prostate cancer." Moreover limitation of each technology are summarized in Table 1.
Comment 4: A diagram would have been very insightful to incorporate in the Manuscript showing the interactions between the protein corona and the nanoparticles.
Answer 4: As requested now a new Figure 1 has been enclosed with this aim. The old Figure 1 now is Figure 2.
Reviewer 2 Report
Comments and Suggestions for Authors
In this paper, the authors have discussed the potential of protein corona on nanoparticles as biomarkers and targeting in prostate cancer. However, prior to publication, the manuscript would benefit from further refinement. Below are specific comments and suggestions for improvement:
Major
1. Please discuss and emphasize the novelty of this review more clearly in the abstract or conclusion section. In other words, mention how this review differs from previous papers in terms of explaining recent updates and the potential of corona nanoparticles in the diagnosis and therapeutics of prostate cancer.
2. The section on main technologies analyzing protein corona-nanoparticles should be slightly expanded.
3. Sections 3 and 4 should focus more on the protein corona rather than primarily on nanoparticles.
4. Could you please elaborate why BPH serves as control groups instead of including healthy donors?
Minor
1. Please check for typos and abbreviations throughout the manuscript. For example, there is a double period in the 10th line of the abstract.
2. Wherever possible, citations should be from recent years (within the last 2–3 years). Ensure that citations are up-to-date and relevant.
Author Response
Dear Reviewer 2, sincerely thank for your comments and suggestions. Now the review is strongly improved. Changes related to Reviewer 2 comments are evidenced in yellow
Minor revisions have been performed as requested.
Comment 1: Please discuss and emphasize the novelty of this review more clearly in the abstract or conclusion section. In other words, mention how this review differs from previous papers in terms of explaining recent updates and the potential of corona nanoparticles in the diagnosis and therapeutics of prostate cancer.
Answer 1: as requested the novelty of the present review has been underlined in the abstract and in paragraph 7
Comment 2: The section on main technologies analyzing protein corona-nanoparticles should be slightly expanded
Answer 2: as requested the section 2 has been expanded with more informations regarding different technologies for protein corona.
Comment 3: Sections 3 and 4 should focus more on the protein corona rather than primarily on nanoparticles.
Answer 3: as requested now section 3 and 4 are more focused on protein corona rather than nanoparticles. In particular: Section 3 and 4: all data for diagnostic application are referred to protein corona and not only on NPs. Data on therapeutic approach are more focused on NPs vehicles for drugs rather tahn protein corona. This aspect is well underlined and explained.
Comment 4: Could you please elaborate why BPH serves as control groups instead of including healthy donors?
Answer 4: this is a relevant point and as requested we now better explain it. In all studies on serum marker for the diagnosis of prostate cancer, the healthy population is that with benign prostatic hyperplasia for the following reasons: 1. it not exists an age-matched population without both PCa and BPH. To compare PCa with healthy men I have to considered males aged less than 30 years with males aged more than 50 years. 2. the goal of an idela Pca marker for diagnosis is to distinguish patients with PCa from those with BPH . This aspect now is better explained in section 6.4.
Reviewer 3 Report
Comments and Suggestions for Authors
In this manuscript, the authors introduce the concept of protein coronas forming on nanoparticles upon interaction with biological fluids. They discuss the mechanisms of protein corona formation and outline detection methods. The manuscript also exemplifies diagnostic and therapeutic applications of protein coronas on nanoparticle surfaces in various oncological models. Additionally, the authors propose an experimental trial design to detect and analyze protein coronas of three differently charged liposomes using plasma samples from patients who are diagnosed with benign prostatic hyperplasia (BPH), non-metastatic prostate cancer (PCa), and metastatic PCa, aiming to develop therapeutic strategies for precision medicine.
Overall, the manuscript provides a valuable overview of the topic. However, certain sections would benefit from further expansion and a stronger focus on protein corona-related content and supporting literature:
1. Please refer to the iThenticate report and revise sentences with high similarity to ensure originality and avoid potential issues with plagiarism.
2. Section 1.1: While the manuscript discusses the definition of protein corona nanoparticles, much of the content focuses on various types of nanoparticles. Shifting the focus more toward introducing protein corona would strengthen the section.
3. Section 1.2: Including a discussion on potential differences in protein corona when nanoparticles are administered via different routes (e.g., oral, inhalation, topical, intramuscular, and intravenous injection) would enhance the manuscript. Additionally, expanding on how nanoparticle size, charge, and shape influence protein corona binding would provide valuable insights.
4. Section 2.3: The statement, “In addition, ultracentrifugation yields false positives or negatives due to dissociation of proteins due to centrifugal forces”, would be better placed in the first paragraph for better flow.
5. Section 3: for reference [12], it would be helpful to expand on why NSCLC and NSCLC-T2DM interact differently with the same NP. This would help readers understand how protein corona formation varies in response to specific comorbidity status.
6. Section 3: the last sentence from the 4th paragraph seems to refer to an incorrect paper. Please review and update the reference.
7. Section 4: The discussion on cellular uptake and the role of the reticuloendothelial system (RES) should be expanded to provide a more comprehensive discussion.
8. Reference 15 and those papers cited in Section 5.1 may have some relevance with NP protein corona or NP protein absorption. However, this connection is not clearly conveyed in the manuscript.
Author Response
Dear Reviewer 3, sincerely thanks for your comments and suggestions. Now ouor review article is strongly improved. Changes in the text related to Reviewer 3 comments are evidenced in green.
Comment 1: Please refer to the iThenticate report and revise sentences with high similarity to ensure originality and avoid potential issues with plagiarism
Answer 1: done, we modify and revise sentences with similarity
Comment 2: Section 1.1: While the manuscript discusses the definition of protein corona nanoparticles, much of the content focuses on various types of nanoparticles. Shifting the focus more toward introducing protein corona would strengthen the section.
Answer2: we focused more on protein corona in section 1.1 . Moreover in section 1.1 we also add Figure 1 as diagram for protein corona interaction.
Comment 3: Section 1.2: Including a discussion on potential differences in protein corona when nanoparticles are administered via different routes (e.g., oral, inhalation, topical, intramuscular, and intravenous injection) would enhance the manuscript. Additionally, expanding on how nanoparticle size, charge, and shape influence protein corona binding would provide valuable insights.
Answer 3:As requested in Section 1.2 we enclosed data and discussion on nanoparticles size, charge and shape influence on protein corona binding and potential differences in protein corona when nanoparticles are administered via different routes.
Comment 4: Section 2.3: The statement, “In addition, ultracentrifugation yields false positives or negatives due to dissociation of proteins due to centrifugal forces”, would be better placed in the first paragraph for better flow.
Answer 4: section 2.3 the statement on ultracentrifugation has been anticipated in the right paragraph.
Comment 5: Section 3: for reference [12], it would be helpful to expand on why NSCLC and NSCLC-T2DM interact differently with the same NP. This would help readers understand how protein corona formation varies in response to specific comorbidity status.
Answer 5:In Section 3 references 12 we show a possible explanation for protein corona composition related to comorbidity associated to NSCLC
Comment 6: Section 3: the last sentence from the 4thparagraph seems to refer to an incorrect paper. Please review and update the reference.
Answer 6: References has been updated in [12]
Comment 7: Section 4: The discussion on cellular uptake and the role of the reticuloendothelial system (RES) should be expanded to provide a more comprehensive discussion.
Answer 7: In Section 4 The explanation on the role of reticoloendothelial system negative influence on NPs drug transortation has been expanded in a more comprehensive discussion
Comment 8: Reference 15 and those papers cited in Section 5.1 may have some relevance with NP protein corona or NP protein absorption. However, this connection is not clearly conveyed in the manuscript.
Answer 8: References 15 has been changed, a new ref 16 enclosed and all other references revised
Round 2
Reviewer 2 Report
Comments and Suggestions for Authors
The manuscript has been improved.
Author Response
Dear Reviewer thanks you for tour commenta that really help us to improve the manuscript
Reviewer 3 Report
Comments and Suggestions for Authors
Most of the questions and comments have been addressed in the revised manuscript. I have a few minor suggestions for further improvement:
1. A reference to Figure 1 should be included in the text.
2. I noticed that the introduction to carbon-related nanoparticles was removed. Since this is mentioned earlier in the manuscript, introducing one or two sentences would enhance continuity.
3. The section introducing hard and soft corona could be improved for clarity. The current structure discusses hard corona first, transitions to soft corona, and then returns to hard corona, which might be confusing to readers. A more streamlined explanation would be beneficial.
4. In the discussion of the RES system, there is a typo "reticoloendothelial". Please check and correct it.
Author Response
Dear Reviewer thank you again for your suggestions.
1. A reference to Figure 1 should be included in the text
Answer: Figure 1 now is cited in the text with reference 5 for the content expressed but the figure was grafically created de novo.
2. I noticed that the introduction to carbon-related nanoparticles was removed. Since this is mentioned earlier in the manuscript, introducing one or two sentences would enhance continuity.
Answer: as requested now two sentences on carbon related NPs have been replaced in introduction section 1.1
3. The section introducing hard and soft corona could be improved for clarity. The current structure discusses hard corona first, transitions to soft corona, and then returns to hard corona, which might be confusing to readers. A more streamlined explanation would be beneficial.
Answer: As requested now the section is more fluid distinguishing between hard and soft coronas
4. In the discussion of the RES system, there is a typo "reticoloendothelial". Please check and correct it.
Answer: the typo has been corrected